# Structure of *Mycobacterium tuberculosis* phosphatidylinositol phosphate synthase reveals mechanism of substrate binding and metal catalysis

Kristīne Grāve [1], Matthew D. Bennett[1] & Martin Högbom [1]

Tuberculosis causes over one million yearly deaths, and drug resistance is rapidly developing. *Mycobacterium tuberculosis* phosphatidylinositol phosphate synthase (PgsA1) is an integral membrane enzyme involved in biosynthesis of inositol-derived phospholipids required for formation of the mycobacterial cell wall, and a potential drug target. Here we present three crystal structures of *M. tuberculosis* PgsA1: in absence of substrates (2.9 Å), in complex with $Mn^{2+}$ and citrate (1.9 Å), and with the CDP-DAG substrate (1.8 Å). The structures reveal atomic details of substrate binding as well as coordination and dynamics of the catalytic metal site. In addition, molecular docking supported by mutagenesis indicate a binding mode for the second substrate, D-*myo*-inositol-3-phosphate. Together, the data describe the structural basis for *M. tuberculosis* phosphatidylinositol phosphate synthesis and suggest a refined general catalytic mechanism—including a substrate-induced carboxylate shift—for Class I CDP-alcohol phosphotransferases, enzymes essential for phospholipid biosynthesis in all domains of life.

[1] Department of Biochemistry and Biophysics, Stockholm University, Svante Arrhenius väg 16 C, SE-106 91 Stockholm, Sweden. Correspondence and requests for materials should be addressed to M.H. (email: hogbom@dbb.su.se)

**M**ycobacterium tuberculosis is the most medically important pathogen of its genus, causing tuberculosis in humans. In 2017 alone 1.6 million deaths and about 10 million new incident cases worldwide were registered. There is a rapid and very alarming development of drug resistance of the disease[1]. Membrane proteins make up targets for the majority of drugs on the market[2,3] and high resolution structures are increasingly important for drug design. Still, at the time of writing there is only one *M. tuberculosis* membrane protein crystal structure available[4,5].

The synthesis machinery of structural and functional components of the mycobacterial plasma membrane and its cell envelope are important targets for the development of novel pharmaceuticals against the pathogen. One such potential target is the mycobacterial phosphatidylinositol phosphate synthase (*M. tuberculosis* PgsA1). PgsA1 catalyzes the formation of phosphatidylinositol phosphate, a precursor of phosphatidylinositol. In turn, phosphatidylinositol is the first building block in the biosynthesis of phosphatidylinositol mannosides and their acylated derivatives, lipomannan, and lipoarabinomannan—structurally complex constituents of the mycobacterial cell wall[6].

The *M. tuberculosis* PgsA1 was identified as a promising candidate for drug development, due to its essential role in growth and proliferation of the pathogen and differences between the Eukaryotic and mycobacterial PI biosynthesis pathways[7–9]. Formation of phosphatidylinositol in mycobacteria is a two-step process. PgsA1 catalyzes the conjugation of the 1' hydroxyl group of D-*myo*-inositol-3-phosphate (ino-P) with a lipid tail of an acceptor substrate, cytidine diphosphate diacylglycerol (CDP-DAG), forming phosphatidylinositol phosphate (PIP) (1). Subsequently, a yet-unidentified phosphatase removes the 3' phosphate from the phosphatidylinositol phosphate, generating phosphatidylinositol (PI)[9–11]

$$D-myo-inositol-3-phosphate + CDP–DAG \rightarrow PIP + CMP$$

(1)

$$PIP \rightarrow PI + P_i$$

(2)

PgsA1 belongs to the class I CDP-alcohol phosphotransferases (CDP-APs), integral membrane proteins that catalyze the formation of a phosphodiester bond to merge a CDP-alcohol and a second alcohol, releasing cytidine monophosphate (CMP). This is an essential step in phospholipid biosynthesis in all domains of life, producing structural phospholipids or their precursors, such as PIP and PI[8,12,13]. Structures of CDP-APs from *Archaeoglobus fulgidus* and *Renibacterium salmoninarum* with different substrate specificity revealed a shared fold where six transmembrane (TM) helices and a long loop structure enclose a large hydrophilic cavity, open towards the cytoplasmic side of the membrane. CDP-APs possess a conserved amino acid sequence motif $D_1xxD_2G_1xxAR...G_2xxxD_3xxxD_4$ in which the four aspartates are believed to coordinate catalytically important divalent metal ions ($Mg^{2+}$, $Mn^{2+}$ or $Co^{2+}$, ref. [14–16] respectively)[17–19].

Despite these structures, information about ligand binding in the family is still limited. To date there are two crystal structures of CDP-APs with the CDP-alcohol substrate bound, but with incomplete catalytic metal sites or at a resolution precluding detailed analysis of substrate binding[17–19]. In turn, any structural information for binding of the second, soluble, substrate is lacking. Together, this significantly hampers structure-based mechanistic proposals, central for scientific understanding and drug design.

Here, we present the crystal structure of the full-length *M. tuberculosis* PgsA1 in absence of metals and substrates (apo) to 2.9 Å resolution and the 1.8 Å resolution structure of the CDP-

DAG complex with a di-nuclear catalytic $Mg^{2+}$ site. To unambiguously assign the metal-binding geometry we also determined the structure of the $Mn^{2+}$—substituted protein to 1.88 Å resolution, allowing X-ray anomalous dispersion identification of the metal binding positions. Together the structures show the details of substrate and metal binding as well as substrate-induced dynamics in protein structure and metal-coordination.

In line with previous studies, co-crystallization with the second substrate, ino-P, was not successful. However, a possible binding site could be identified based on the position of a serendipitously bound Mn-citrate complex in the metal substituted crystal structure. Docking studies with ino-P in combination with mutagenesis and activity assays support this assignment and provide a plausible binding mode for this substrate. Based on the combined results we propose a structure-based mechanism for phosphatidylinositol phosphate synthesis in *M. tuberculosis*.

## Results

**Crystal structure of *M. tuberculosis* PgsA1 in the apo and metal-substituted forms: metal site structure and dynamics of metal position 2.** PgsA1 crystallizes as a homodimer—with two protein molecules per asymmetric unit, with an overall fold resembling previously reported structures of CDP-alcohol phosphotransferases[17–19]. The dimer interface is formed mostly by TM helices 3 and 4 contributing hydrophobic interactions between the protomers (Fig. 1a). In the apo structure the first 16 residues in chain A, and 15 in chain B, are disordered. In the metal-free form, the metal binding residues D68 and D71 appear to be almost completely disordered while D89 and D93 have better-defined electron density. Additionally, residues 149–151 (-FIE-), located on a loop connecting transmembrane helices 4 and 5, and eight C-terminal residues could not be traced in the electron density. The electron density map does not contain any unexplained positive density in the vicinity of the protein active site that could correspond to the substrates or metal ions of PgsA1, therefore this structure is assigned to be in the apo state. Positive electron density is found at the interface between the two protein molecules in the asymmetric unit, close to the R83 in chain A and the R115 of chain B possibly corresponding to a low occupancy deoxycytidine triphosphate molecule from the crystallization condition.

Interestingly, TM 2 shows a $3_{10}$-helix motif—a main chain hydrogen bond between L66 and M69, flanking two aspartates of the conserved CDP-AP motif (Supplementary Fig. 1). This short $3_{10}$ motif is present also in the other CDP-AP protein structures determined to date[17–19], irrespective of the presence or absence of bound ligands in the active site and appears to be a common feature of the class.

In order to define the metal ion binding sites, $Mg^{2+}$ was substituted with $Mn^{2+}$, a divalent cation that also supports catalysis in CDP-APs, including *M. tuberculosis* PgsA1[15,20]. $Mn^{2+}$ has very similar electronegativity and ionic radius as $Mg^{2+}$ but allows anomalous dispersion measurements to unambiguously determine the metal binding positions. The $Mn^{2+}$ substituted crystals diffracted to 1.88 Å and belonged to space group $P2_12_12_1$ with one homodimer per asymmetric unit as for the apo structure. It was shown previously that the aspartates of the signature motif are involved in coordination of a di-nuclear metal site in CDP-APs[19]. As expected, our metal-substituted structure reveals a di-nuclear metal binding site coordinated by D68, D71, D89, and D93 of the conserved sequence motif, as indicated by two clear peaks within a metal coordinating distance to the chelating carboxylate side chains (Fig. 1b, c). In chain B of the *M. tuberculosis* PgsA1 homodimer, however, the anomalous difference density, corresponding to the aspartate-chelated

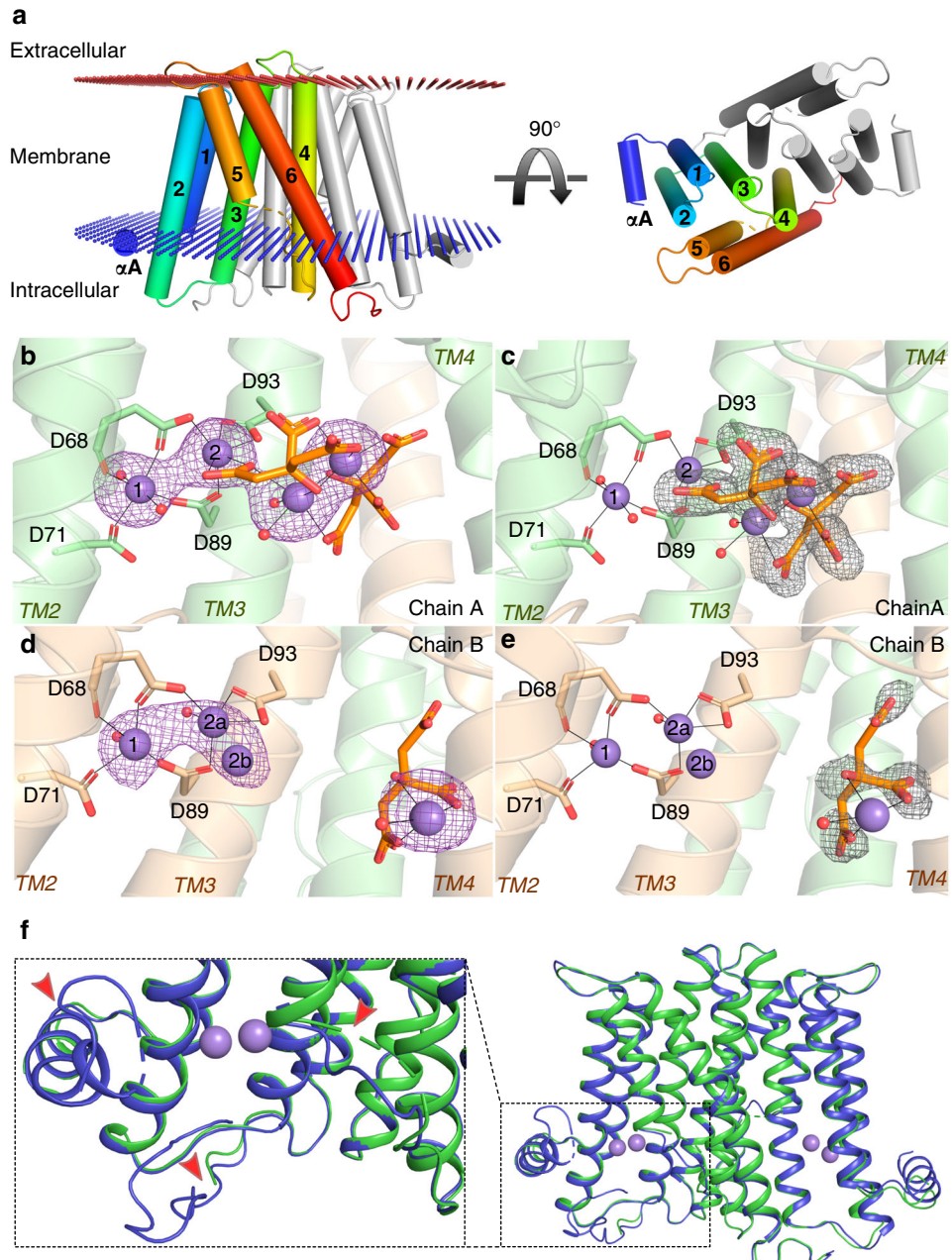

**Fig. 1** Architecture of apo and Mn-substituted *M. tuberculosis* PgsA1. **a** Transmembrane helices are numbered (1–6) and colored in rainbow, blue—N terminus, red—C terminus. **b**–**e** Metal binding sites in the PgsA1 dimer probed by Mn-substitution and anomalous scattering difference maps. Metal binding site 2 in chain B show dynamics with two possible metal binding positions suggesting metal ion mobility depending on ligand binding in the proximal hydrophilic pocket. Anomalous difference density around Mn ions is contoured at 4 σ and shown as a purple mesh. $F_o - F_c$ omit electron density map for the bound citrates (in orange sticks) is contoured at 3.0 σ and shown as a gray mesh. Black lines denote metallic bonds. **f** Structural differences between apo (blue) and Mn-containing (green) structures are indicated with red arrows. Similar structural flexibility is observed in the second protomer of the dimer. Citrate and citrate-chelated metal ions are omitted for clarity. Manganese ions are shown as purple spheres, solvent molecules—as smaller red spheres

Mn-ions, is extended indicating two alternate binding positions for the metal ion in site 2 (indicated as 2a and 2b in Fig. 1d, e). Besides the anticipated protein-bound metal ions, additional anomalous difference density peaks were found: two in chain A and one in chain B. These additional Mn-ions appeared to be coordinated by serendipitously bound citrate molecules, most likely derived from the crystallization condition. Two citrate molecules are chelating two Mn-ions in chain A and one citrate molecule is found chelating one Mn ion in chain B (Fig. 1b–e). The citrate-Mn complex is most likely not biologically relevant and result from the presence of 100 mM citrate and 120 mM

$Mn^{2+}$ in the crystallization solution. The different binding in the two protomers in the asymmetric unit is likely due to differences in accessibility and/or structural asymmetry due to crystal packing. Most likely, the same effects induce the observed differences in coordination for the catalytic dinuclear metal site observed between protomers.

The presence of the metal ions and bound Mn-citrate ligand complexes in both protein chains appear to have a stabilizing effect on the N-terminal helix and C-terminal loop which now can be indubitably traced in the electron density (Fig. 1f). One of the citrate molecules binds to approximately the same area of the

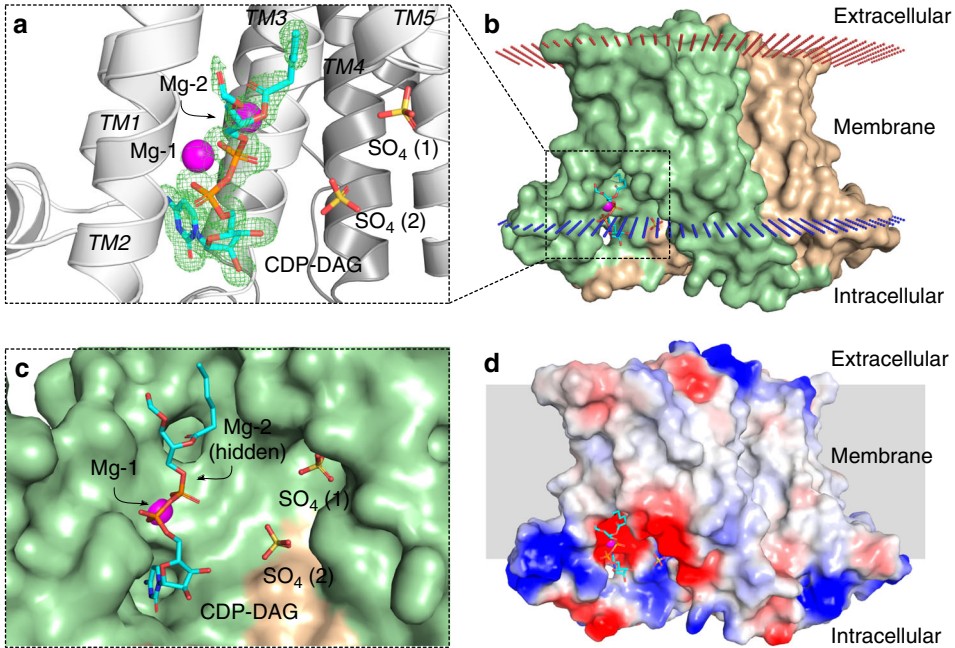

**Fig. 2** Overview of CDP-DAG binding to *M. tuberculosis* PgsA1. **a** Both protomers of the *M. tuberculosis* PgsA1 contain bound CDP-DAG (in sticks, cyan). $F_o - F_c$ omit electron density map for the CDP-DAG is contoured at $2.5\,\sigma$ and shown as green mesh. Magnesium ions are shown as magenta spheres. Sulfate molecules are shown as yellow/red sticks. **b** Calculated spatial position of *M. tuberculosis* PgsA1 in a lipid bilayer[21]. **c** Surface representation of the CDP-DAG binding cavity. **d** Surface representation of *M. tuberculosis* PgsA1 in the same orientation as in (**b**). Surface potential is calculated using PyMOL 2.0.0[22]

positively charged pocket in both protein chains. In chain A an additional citrate molecule is found, forming an $(Mn)_2$-citrate$_2$ complex, also coordinating the catalytic metal site. The $(Mn)_2$-citrate$_2$ complex appears to be stabilized by the ordered C-terminal loop of chain B, protruding into the hydrophilic pocket of chain A. The final 6 residues of the ordered loop (*-ENLYFQ*) is a remnant from TEV protease digestion. Chain A could be traced to residue A210, while the last 7 C-terminal residues could not be modeled.

***M. tuberculosis* PgsA1 in complex with $Mg^{2+}$ and CDP-DAG: details of substrate interaction and carboxylate shift dynamics of metal binding**. Co-crystallization of the PgsA1 protein with CDP-DAG and $Mg^{2+}$ in lipidic cubic phase yielded crystals diffracting to 1.8 Å and belonging to space group $P2_12_12$. The CDP-DAG binding site is established by the four aspartates of the conserved signature motif ($D_1$–$D_4$), primarily involved in coordination of two magnesium ions, Mg-1 and Mg-2, which in turn coordinate the CDP-DAG phosphates. The nucleotide moiety of the substrate is bound into a cleft formed by TM helices 1–3, lined by G72, A75, and G85 and exposed to solvent (Figs. 2 and 3). Interestingly, the nucleotide group of the CDP-DAG shares only a few relatively weak hydrogen bonds with the protein, specifically with residues D31 and T34 located on TM1 and the backbone amine of T82 located on the flexible loop. Additionally, G72 and R76 are involved in coordination of the α-phosphoryl group (Fig. 3). The long acyl chains of the substrate are disordered, suggesting high flexibility. During model building, these acyl chains were truncated in both CDP-DAG molecules (Figs. 2 and 3). The position of the glycerol backbone and remnant density, however, suggests that they protrude from the protein surface made up of TM helices 2 and 5.

The di-$Mg^{2+}$ site is intact in the CDP-DAG complex structure. Mg-1, coordinated by the N-terminal aspartates of the signature motif, establishes the binding site by also coordinating the substrate phosphates. There are however interesting differences

between the protein chains, illustrated in Fig. 3. The metal site in chain A replicates the coordination and metal binding sites 1 and 2a observed in the $Mn^{2+}$ substituted protein (Fig. 1d, e). The two $Mg^{2+}$ ions are 4.6 Å apart; we denote this coordination mode as the "tight" state. In chain B, on the other hand, the metal site adopts the other state observed in the $Mn^{2+}$ substituted protein, with the metal ions occupying positions 1 and 2b. In this "relaxed" state, the metal-metal distance is 5.4 Å. This difference is primarily achieved because of a carboxylate shift of the Mg-2 coordinating D89 and the presumed catalytic base D93 (ref. [17,18]). Together, this suggests that the metal in site 2 has two possible binding positions and that the coordination mode is influenced by the surrounding environment. Both $Mg^{2+}$ ions in chain A are 6-coordinate, Mg-1 is coordinated by D68, D71, D89, and the CDP-DAG phosphate groups. Mg-2 is coordinated by D89, which adopts two conformations as supported by the electron density—in chain A, bridging both metal ions and in chain B coordinating only Mg-2. Water molecules complete the metal primary coordination spheres (Fig. 3).

There are two pronounced anionic membrane phospholipid binding sites in the deep hydrophobic cleft between two protein chains on both sides of the protein dimer. These hydrophobic clefts extend in the direction of the extracellular side of the protein towards the positively charged residue patch. This patch is formed by R115 from one protomer together with W106 and H111 from the other. These phospholipid-binding sites are especially well defined in the CDP-DAG bound structure. The side chains of R115, W106, and H111 are within hydrogen bonding distance to a bound $SO_4^{2-}$ ion. This sulfate ion, presumably originating from the crystallization solution, potentially indicates the binding position for the negatively charged head group of a phospholipid. Based on the shape of the corresponding electron density and surroundings, the unidentified lipids are modeled as monoolein fragments (2,3-dihydroxypropyl fatty acids) with varying chain length (Supplementary Fig. 2), as monoolein was used for the formation of the lipidic

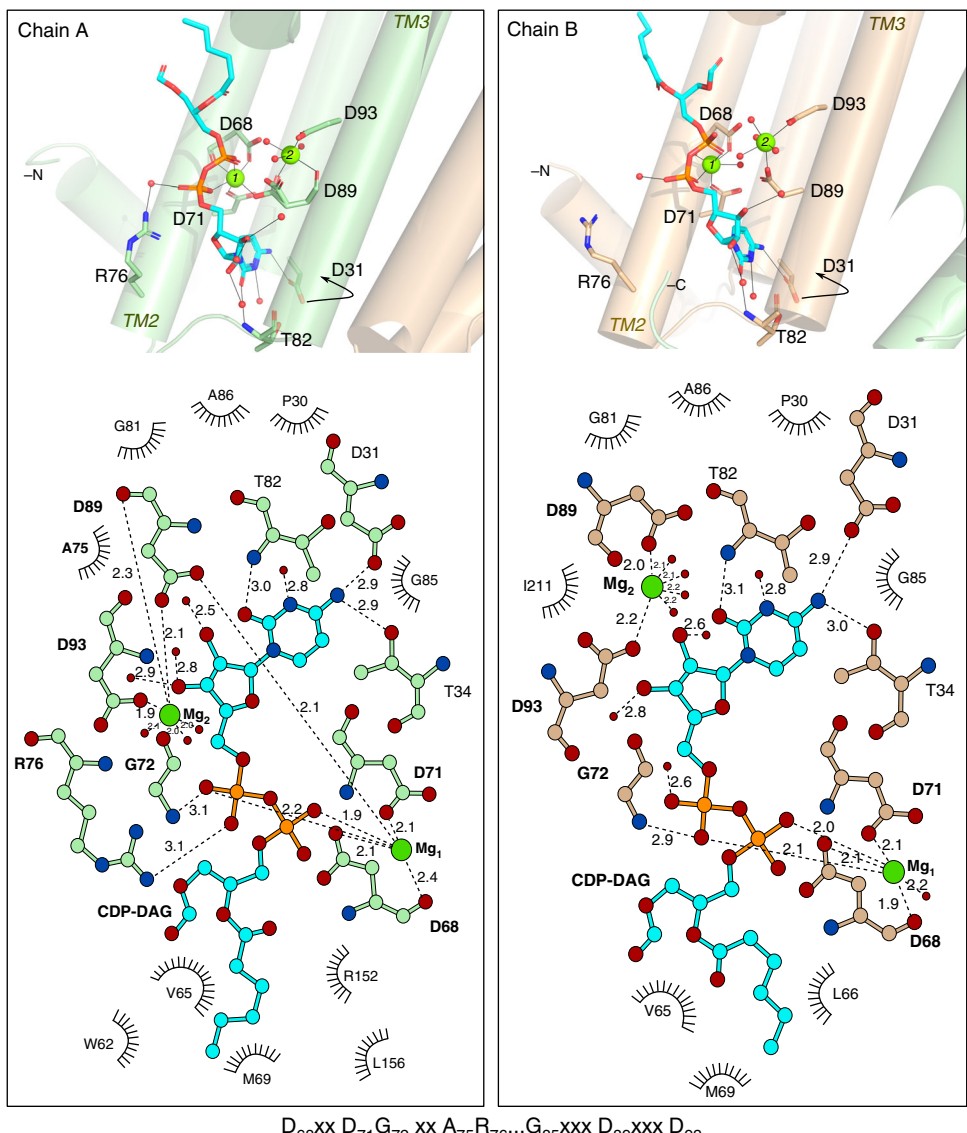

$D_{68}xx\ D_{71}G_{72}\ xx\ A_{75}R_{76}...G_{85}xxx\ D_{89}xxx\ D_{93}$

**Fig. 3** Comparison of CDP-DAG and metal ion coordination in the two protein chains of the homodimer. Chain A ("tight") and Chain B ("relaxed") metal coordination positions corresponding to $Mg_2$ being located in the a and b binding sites respectively (see also Fig. 1e). CDP-DAG is shown in cyan. Relevant hydrogen bonds are shown as dashed lines and their distances are expressed in Å. Solvent molecules are shown as smaller red spheres, Mg ions—as green spheres. The signature motif of the CDP-alcohol phosphotransferases in the bottom panel is shown with *M. tuberculosis* PgsA1 residue numbering

mesophase for LCP crystallization. Two additional sulfate ions in close vicinity to the site could also be modeled in this structure, yet at somewhat lower occupancies.

**D-*myo*-inositol-3-phosphate binding site probed by molecular docking**. A hydrophilic pocket, exposed to the cytoplasm, is formed by both protein chains and is lined by flexible positively charged residues—R94, R152, R155, R195, and K135, as well as R137 of the second protein chain (Fig. 4a). This has previously been speculated to be the position for binding of the alcohol substrate in CDP-alcohol phosphotransferases[18,19]. In all crystal structures, except for the Mn-citrate bound, at least one sulfate ion ($SO_4^{2-}$ (1)) is bound in the apical side of the pocket, hydrogen bonded to S132, R155, R195 and the main chain of R152. In the Mn-citrate complex structure one of the citrate carboxylates assumes the position of $SO_4^{2-}$ (1) in both protein chains (Fig. 4b, c). It appears plausible that this conserved positively charged binding site is involved in coordinating the phosphate moiety of the ino-P substrate. In addition, there is a second bound sulfate ion ($SO_4^{2-}$ (2)), seen only

in the CDP-DAG bound structure and located about 5 Å away from the catalytic aspartate D93 (Fig. 4a). It is coordinated by K135 and R137 of the second protein chain located on the cytosolic side of the protein, close to the calculated position of the membrane plane (Figs. 2b and 4a). In the Mn-citrate complex structures this binding position is also occupied by a citrate carboxylate in chain A, while in chain B the corresponding carboxylate of the citrate is slightly shifted (Fig. 4b, c).

Co-crystallization and soaking attempts with ino-P proved unsuccessful in line with the low binding affinity for this substrate[19] and likely competition for the binding site by the 100 mM sulfate present in the crystallization condition. However, the serendipitously obtained di-manganese citrate complex, as well as the bound sulfate molecules, suggests that this solvent-accessible pocket proximal to the CDP-DAG binding site is a plausible binding site for the second substrate.

We proceeded with molecular docking to probe if this binding site would support a plausible binding mode for ino-P. Docking of ino-P was performed against all three structural models: apo,

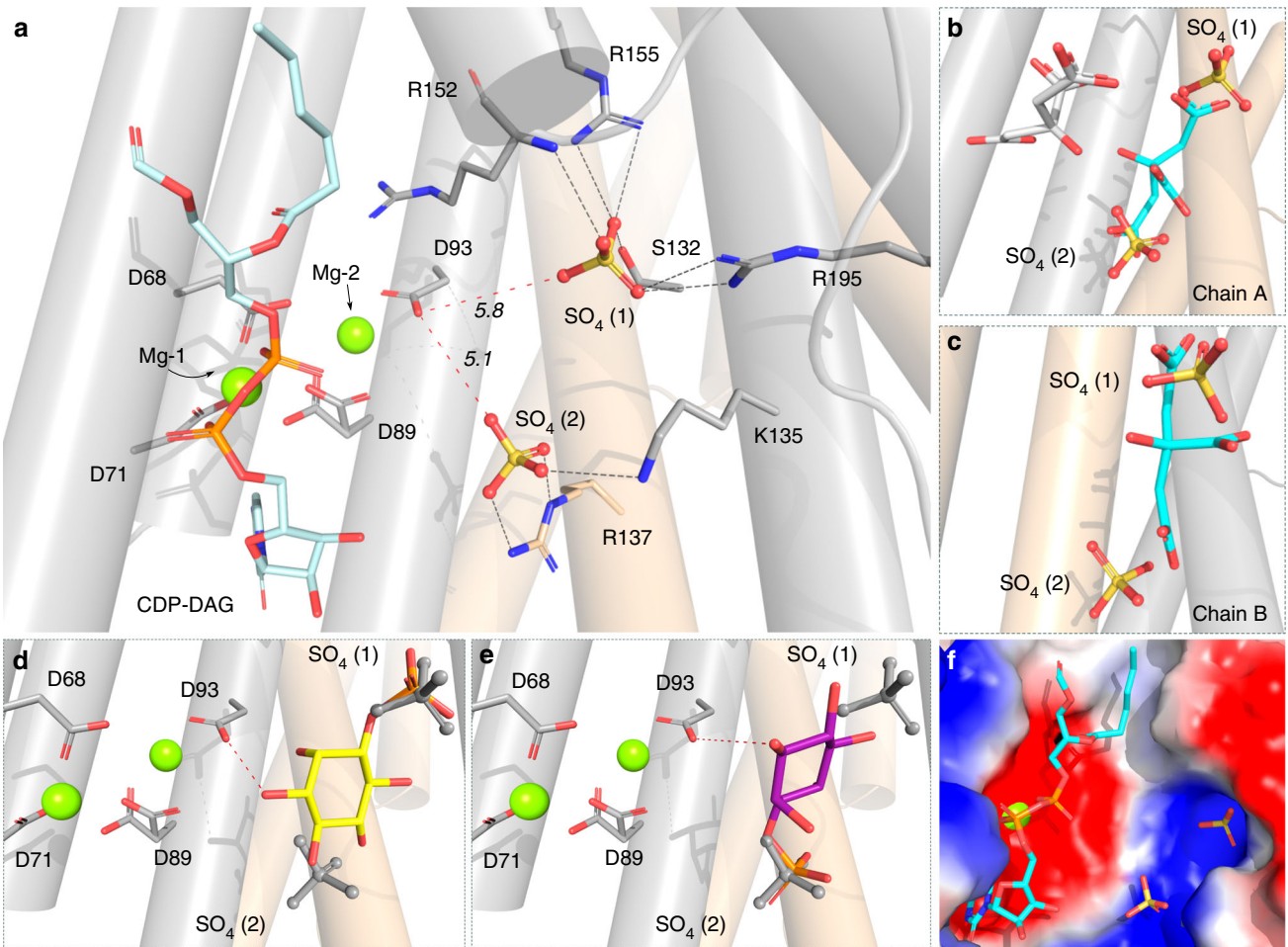

**Fig. 4 Conserved positively charged sulfate and citrate binding pocket in proximity to the metal site indicates the binding site for D-*myo*-inositol-3-phosphate. a** Localization and coordination environment of two sulfate ions in the CDP-DAG bound *M. tuberculosis* PgsA1 at the active site. Black dashed lines indicate hydrogen bonds between the sulfate ions and protein residues. Red dashed lines denote sulfate ion proximity (in Å) to the catalytic aspartate and the metal site. **b**, **c** Structural overlay of sulfate ion binding site in the CDP-DAG-bound *M. tuberculosis* PgsA1 with citrate ion binding site in the Mn-citrate *M. tuberculosis* PgsA1 in both protein chains, A and B. Citrate-chelated metals are omitted for clarity. The citrate molecule, which binds in the same site as the sulfate ions, is shown in cyan. The second citrate found only in chain A is shown in silver sticks. **d**, **e** Two representative binding poses of D-*myo*-inositol-3-phosphate were selected from two preeminent molecular docking clusters. The top-scored binding pose, denoted #1 in the main text, is shown in yellow sticks; and the second potential binding pose, denoted #2, is shown in magenta sticks. Red dashed line denotes proximity of the D-*myo*-inositol-3-phosphate 1′ hydroxyl group and the catalytic aspartate. The sulfate ions of the *M. tuberculosis* PgsA1 structure are shown in gray sticks. Bound CDP-DAG is omitted for clarity. **f** The sulfates are bound to the positively charged pocket (blue) in the vicinity to the metal site and the catalytic aspartate. Surface potential is calculated using PyMOL 2.0.0[22]. CDP-DAG is shown in cyan. Mg ion protruding through the surface representation is shown in green

CDP-DAG and Mn-citrate bound with the ligands removed. Two separate clusters of plausible binding modes were obtained in the docking experiments. In both cases, the phosphate of the Ino-P substrate overlaid well with the sulfate ions observed in the CDP-DAG structure. Both selected binding modes resulted in a positioning of the inositol headgroup that placed the 1′ hydroxyl within hydrogen bonding distance of the Mg-2 coordinating fourth aspartate (D93) in the signature motif, consistent with bond formation to the correct hydroxyl of the substrate (Fig. 4d, e). Binding mode #1 (yellow) placed the phosphate at the position of sulfate binding site 1 which appears conserved in all previously determined structures of CDP-alcohol phosphotransferases while binding mode #2 (magenta) placed the phosphate at the position of the second sulfate binding site, observed in CDP-DAG bound *M. tuberculosis* PgsA1. The best binding energy was observed for binding mode #1 which was also the top solution in docking against both the CDP-DAG and the Mn-citrate complex structures. Both of the plausible binding modes were also observed when docking against the apo structure, albeit with

worse binding energies and not representing the top solution. In all cases, binding mode #1 was predicted to be better than binding mode #2. The docking results are consistent with the experimental results that CDP-DAG promotes binding of ino-P (ref. [19]) and shows that this effect is at least partly mediated via protein conformational changes.

**Mutagenesis supports the inositol phosphate binding mode suggested by molecular docking.** In order to validate the results obtained by molecular docking, several previously uncharacterized mutations, A90Y, R94K/Q, Y133F/E, and R137K/Q, were introduced and protein specific activity was measured using a malachite green based discontinuous assay in proteoliposomes. None of these four residues are part of the signature motif, however, the residue in sequence position 94 is always positively charged (K or R) (Supplementary Fig. 3). Based on our crystal structures and molecular docking results, we expected these mutations to affect protein activity negatively due to

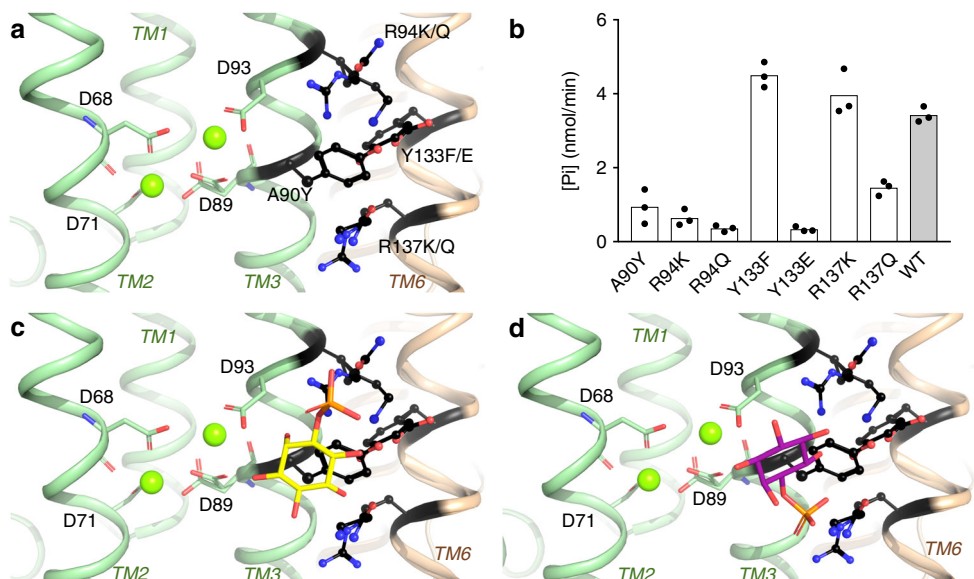

**Fig. 5** Characterization of *M. tuberculosis* PgsA1 point mutants. **a** Graphic representation of mutant residue localization relative to the metal site. Note, that Y133 and R137 are part of the second protein chain. **b** Activity of the point mutants is shown in comparison to wild-type (WT) *M. tuberculosis* PgsA1 in presence of D-*myo*-inositol-3-phosphate. The activity assay was performed in proteoliposomes; data for 180 min measurements for $n = 3$ are shown. **c**, **d** Two representative D-*myo*-inositol-3-phosphate binding poses from two top-scoring docking model clusters. The top-scored binding pose is shown in yellow sticks; and the second most plausible binding pose—in magenta sticks

compromised binding of the ino-P caused by altered local positive surface charge or steric interference with the binding of the substrate. All the mutants were expressed and purified similarly to the wild-type (WT) PgsA1 and reconstituted into proteoliposomes. The activity was assayed in triplicate and presented in Fig. 5b.

Docking results indicate two potential binding modes of the ino-P in the pocket with the phosphate group occupying the $SO_4^{2-}$ binding positions (1) and (2), respectively (Figs. 4d, e and 5). As anticipated, when A90 is exchanged for a bulkier side chain, tyrosine, activity is significantly reduced, presumably due to steric hindrance and substrate positioning of the inositol ring relative to D93 and the metal site. This mutation would be expected to influence both potential binding modes in a similar way. However, the mutational studies also show that R94 is a key residue for activity. Its conservative substitution for lysine diminishes protein activity dramatically, and substitution with glutamine nearly abolishes activity. R94 thus appears to be critical for ino-P binding. R94 contributes to the $SO_4^{2-}$ (1) binding site, directly interacting with the Ino-P phosphate in binding mode #1 (Fig. 5). On the other hand, R137, located on the second protomer, contributes to the $SO_4^{2-}$ (2) binding site and interacts with the Ino-P phosphate in binding mode #2 (Figs. 4a and 5). Interestingly, the R137K mutation does not have a similar effect on activity as the R94K, in this case protein activity is comparable to wild-type. A R137Q mutation, however, reduces activity by approximately half. Y133 from the second protomer also lines the substrate-binding pocket. The Y133F mutant is almost equally active as the wild-type, thus removal of the hydroxyl group does not prevent productive substrate binding. Substitution of this tyrosine to glutamate, however, completely abolishes the activity, presumably due to disrupting the charge distribution in $SO_4^{2-}$ binding position (1). Previous studies show that affinity for ino-P in the P153V mutant is reduced[19]. P153 stacks against $SO_4^{2-}$ (1) in the crystal structures and if substituted to valine, would interfere with the binding of the phosphate group of the ino-P in binding mode #1. Together, the docking and mutational data suggest that binding mode #1 with the phosphate group of the

inositol occupying the position of the $SO_4^{2-}$ (1) is the most plausible and represents a good model for binding of ino-P in PgsA1.

## Discussion

*M. tuberculosis* PgsA1 has been identified as a promising candidate for drug development because of its vital role in growth and proliferation of the pathogen as well as the differences between eukaryotic and mycobacterial PI biosynthesis pathways[8,9]. In recent years a number of crystal structures of class I CDP-APs were released, shedding light on overall architecture and functional aspects of conserved residues in these enzymes. Based on crystal structures of the bifunctional enzyme DIPPS[17] and AF2299[18] from the hyperthermophylic archaeon *Archaeoglobus fulgidus* and RsPIPS from the Gram-positive bacterium *Renibacterium salmoninarum*[19] catalytic mechanism of CDP-APs have been proposed. However, the limited resolution of substrate-bound structures (3.6 Å for the CDP-DAG bound structure of RsPIPS)[19], as well as the lack of structural information of the catalytic di-nuclear metal site, has significantly hampered detailed mechanistic proposals, central for scientific understanding and drug design.

In line with previous studies, co-crystallization with D-*myo*-inositol-3-phosphate was not successful. However, the Mn-citrate complex revealed a plausible binding pocket for this substrate (Fig. 4b, c). This presumption is strengthened by the presence of a conserved sulfate binding site ($SO_4^{2-}$ (1), Fig. 4a). Moreover, biochemical data suggest that CDP-DAG primes the active site for the binding of ino-P[19], the docking results are also consistent with this observation. Previous mutational studies as well as the ones reported here support the proposed binding mode #1 for ino-P. First, functional characterization of the strictly conserved R155 and R195 coordinating the structurally conserved $SO_4^{2-}$ ($SO_4^{2-}$ (1)), by Clarke et al. in addition to the observed inhibitory effect of $SO_4^{2-}$ and $PO_4^{2-}$ on enzymatic activity, suggest a critical importance of these two side chains in specific binding of the phosphate group of the ino-P[19]. The docking experiments suggest

that R155 and R195 are involved in coordinating and correctly orienting the ino-P with its 1' hydroxyl relative to the catalytic site (Fig. 4a, d and e). In addition, sequence alignment of CDP-diacylglycerol-inositol 3-phosphatidyltransferases show that R94 is conserved in CDP-AP enzymes utilizing ino-P as a substrate. R94 is located close to the estimated position of the ino-P phosphate group (for the binding mode #1). Our mutagenesis experiments show that R94 is crucial for protein activity. The R94 residue is much more sensitive to mutational changes than R137, interacting with the ino-P phosphate in binding mode #2 (Fig. 4). The K135 side chain, which is coordinating the $SO_4^{2-}$ (2) (Fig. 4a), is also less sensitive to mutations, indicating that ino-P binding mode #2 is less likely[19]. Additional mutations designed to alter the charge of the $SO_4^{2-}$ -housing pocket or to ensure steric obstruction for binding of the ino-P in this site, led to compromised or abolished enzymatic activity, in agreement with the suggested ino-P binding mode.

The CDP-DAG bound *M. tuberculosis* PgsA1 structure reveal an outward movement and rotation of TM2 along its axis and subsequent slide of αA along the TM2, as compared to the apo structure (Supplementary Fig. 4 and Movie 1 and 2). Notably, the most prominent movement of TM2 occurs in the $3_{10}$-helix motif flanked by the conserved aspartates 68 and 71 (Supplementary Fig. 1). Since this $3_{10}$ motif is present in all to date published structures of CDP-alcohol phosphotransferases, it seems to have a functional role. Most likely, outward movement of TM2 allows accommodation of bulky lipid substrates such as CDP-DAG in this class of enzymes. M69 appears to play a crucial role in in this process: Clarke el al. showed that M69A exhibits a significant increase in enzymatic activity, while nearly no activity was observed in the M69W variant[19]. These results are well in agreement with the structural data, as M69 appears to regulate the access of the large hydrophobic substrate to the active site. Substitution to a smaller side chain, such as alanine, potentially facilitates CDP-DAG binding while a bulky side chain, such as tryptophan, on the other hand—would restrict it.

Interestingly, our metal-bound structures of *M. tuberculosis* PgsA1 reveal pronounced mobility of the metal bound in site 2 (coordinated by D89 and D93). This carboxylate shift, concomitant with a change in coordination mode and binding position of the metal, appears to be induced by ligand binging in the presumed ino-P binding site.

Clarke et al. found that *M. tuberculosis* PgsA1 is more active when CDP-DAG binds prior to D-*myo*-inositol-3-phosphate[19], and most likely primes the di-nuclear metal site and catalytic residues for the binding of the second substrate, thus helping to orient it correctly prior to catalysis. The presence of a mobile region on TM2 (the $3_{10}$ motif), the dynamics of the metal site and its surroundings is most likely characteristic features of these enzymes. The Mn-citrate and the CDP-DAG bound *M. tuberculosis* PgsA1 structures reveal two positions and coordination modes for the site 2 metal ion (denoted "tight" and "relaxed" states, respectively). This movement is accompanied by carboxylate shifts of D89 and D93—the suggested catalytic base. We hypothesize that the carboxylate shift of the third and fourth aspartates of the signature motif, induced by binding of the second alcohol substrate (CDP-DAG), primes the site for proton abstraction and nucleophilic attack in class I CDP-APs (Fig. 6).

## Methods

### Construct design and cloning of wild-type *M. tuberculosis* PgsA1 and mutants.
*Rv2612c* gene (Gene ID: 888209; UniProt ID: P9WPG7) was amplified from *Mycobacterium tuberculosis* H37Rv gDNA and cloned into pET_cLIC_GFP vector using ligation-independent cloning (LIC)[23,24]. In short, the LIC procedure included vector linearization at a unique *SwaI* site in the middle of the LIC cassette. Using the 3′ to 5′ exonuclease activity of T4 DNA polymerase, single-stranded overhangs are generated on the digested vector and generated PCR products. Exonuclease

treatment in presence of specific dNTP allows generation of stable complimentary overhangs on the PCR products and vector. Then the treated vector and insert were mixed in a 1:3 ratio, respectively and transformed into *E. coli* after a short incubation.

Expression of the resulting PgsA1 and C-terminal folding indicator GFP[25] fusion protein followed by His$_{10}$ purification tag was controlled by T7 promoter. Additionally, a TEV-specific cleavage site was included, allowing removal of GFP-His$_{10}$ in the final stage of the protein purification.

A90Y, R94K, R94Q, Y133F, Y133E, R137K, and R137Q point mutants were prepared using a QuikChange Lightning (Agilent, CA, USA) site-directed mutagenesis kit following the instructions of the manufacturer and using wild-type *Rv2612c* construct in pET_cLIC_GFP vector as the PCR template. Presence of desired mutations was confirmed by sequencing. All primer sequences used in this study are listed in Supplementary Table 1.

### Expression and purification of wild-type *M. tuberculosis* PgsA1 and mutants.
Wild-type and mutant PgsA1 were expressed overnight at 20 °C in Rosetta 2 (DE3) cells after induction with 1 mM isopropyl β-D-1-thiogalactopyranoside (IPTG) at OD$_{600}$ 0.6 in 1× M9 minimal salts medium (Formedium, Norfolk, UK) supplemented with 0.5 % D-glucose (w/v), 2 mM MgSO$_4$, 0.1 mM CaCl$_2$, 50 μg per mL carbenicillin and 25 μg per mL chloramphenicol. Cells were harvested by centrifugation at 5000 × *g* at 4 °C for 15 min and disrupted in buffer L containing 20 mM Tris-HCl pH 7.4, 300 mM NaCl, 2 mM 2-mercapthoethanol (β-Me), 10 μg per mL DNase I and EDTA-free antiprotease mixture tablets (Roche, Basel, Switzerland) using EmulsiFlex-C3 system (AVESTIN, Ottawa, Canada). Cell debris was removed by centrifugation at 15,000 × *g* at 4 °C for 15 min. Cell membranes were pelleted by ultracentrifugation for 1 h at 138,000 × *g* at 4 °C. Membrane pellets were then resuspended in buffer R containing 20 mM Tris-HCl pH 7.4, 300 mM sucrose, 200 mM NaCl. Solubilization of membranes was performed for 2.5 h at 4 °C on a rocking platform after addition of 1% (w/v) *n*-dodecyl-β-d-maltopyranoside (DDM, Anatrace, OH, USA) in 50 mM Tris-HCl pH 7.4, 200 mM NaCl, 10% glycerol (v/v) and 10 mM imidazole followed by ultracentrifugation for 1 h at 138,000 × *g* at 4 °C for the removal of any unsolubilized material. Proteins were purified from the supernatant using immobilized metal-affinity chromatography (Ni-NTA, Qiagen). The solubilized membranes were incubated with pre-equilibrated Ni-NTA beads (0.7 ml for 40 ml soluble fraction) overnight. The beads were washed with 8 column volumes of 20 mM Tris-HCl pH 7.4, 300 mM NaCl, 0.04% (w/v) DDM and 100 mM imidazole (pH 7.5). The protein was then eluted from the beads with 5 column volumes of 20 mM Tris-HCl pH 7.4, 300 mM NaCl, 0.04% (w/v) DDM, 5% glycerol and 450 mM imidazole pH 7.4. To remove imidazole, the elution fraction was concentrated using 100,000 MWCO concentrator (Vivaspin, MA, USA) and loaded onto Superdex 200 16/60 (GE Healthcare) in 20 mM Tris-HCl pH 7.4, 200 mM NaCl, 0.04% (w/v) DDM, 5% glycerol (v/v) and 2 mM β-Me. GFP-His$_{10}$-tag was further removed by overnight incubation at 4 °C with equimolar concentration of TEV protease. The next day, a reverse Ni-NTA step was performed in order to remove any uncleaved protein, TEV protease, and GFP-His$_{10}$ tag. As a final polishing step, flow-through from previous step was concentrated using 100,000 MWCO concentrator and loaded onto Superdex 200 16/60 in 20 mM Tris-HCl pH 7.4, 100 mM NaCl, 0.02% (w/v) DDM, 5% glycerol (v/v) and 1 mM tris(2-carboxyethyl)phosphine hydrochloride (TCEP-HCl). WT and mutant PgsA1 protein eluted as a sharp monodisperse peak.

### Protein crystallization.
Crystals of *M. tuberculosis* PgsA1 were grown in lipidic cubic phase (LCP)[26,27] at 22 °C. Molten monoolein (Hampton Research, CA, USA) was mixed with the concentrated protein (12–15 mg per mL) in a 3:2 ratio, respectively, using gas-tight coupled syringes. Protein drops containing 50 nL of monoolein-protein mixture and 800 nL of precipitant were set up using Mosquito LCP robot (TTP Labtech, Melbourn, UK) in a glass sandwich plates (Molecular Dimensions, Suffolk, UK). Apo protein crystals were grown in 33% PEG 400 (v/v), 0.1 M NaCl, 0.1 M MgSO$_4$, 0.1 M trisodium citrate dihydrate pH 6 and dCTP as an additive at 4–6 °C. Mn-citrate bound PgsA1 crystals were grown in 31% PEG 400 (v/v), 0.1 M NaCl, 0.12 M MnCl$_2$ and 0.1 M trisodium citrate dihydrate pH 6. Crystals grew within 2–3 weeks at 20 °C. However, as crystals grown in this condition have a citrate molecule bound potentially interfering with substrate binding and the protein requires Mg for the activity[19], crystallization conditions were modified for the growth of CDP-DAG bound PgsA1. In this case monoolein was doped with 4.5% (m/m) CDP-DAG (18:1 CDP-DG 1,2-dioleyl-sn-glycero-3-citidine diphosphate, Avanti Polar Lipids, AL, USA) and mixed with the protein (12–15 mg per mL). Protein drops then set up as described above. CDP-DAG bound crystals grew in 33% PEG 400 (v/v), 0.1 M NaCl, 0.1 M Bis-Tris pH 6, and 0.1 M MgSO$_4$ for 2–3 weeks. Crystals were harvested using MicroMounts (MiTe-Gen, NY, USA) and flash cooled directly in liquid nitrogen without additional cryoprotection.

### Diffraction data collection.
Diffraction data for the apo and Mn-citrate complex *M. tuberculosis* PgsA1 were collected at 100 K on beamlines Xo6SA (PXI) at the SLS (Paul Scherrer Institute, Villigen, Switzerland) at 0.99 Å and 1.00 Å wavelength, respectively. The near-Mn absorption edge high-redundancy anomalous dataset (1.88 Å wavelength) was collected prior to the native dataset. The CDP-

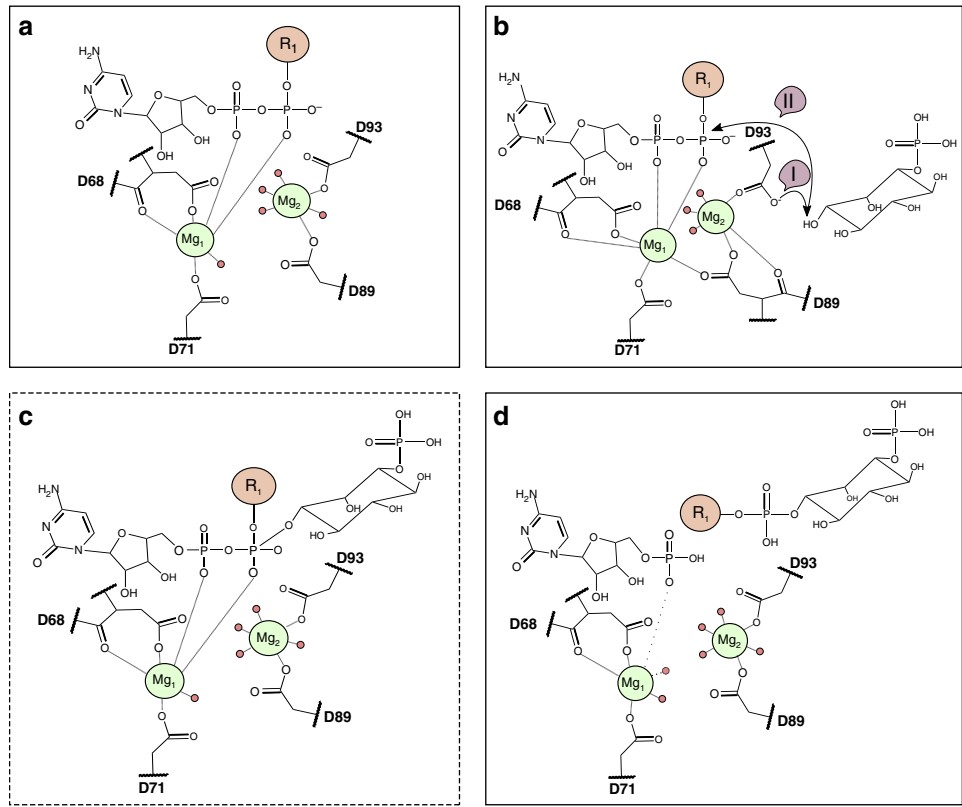

**Fig. 6** Proposed model for substrate-induced initiation of catalysis. **a** Resting "relaxed" state with CDP-DAG bound. **b** Binding of D-*myo*-inositol-3-phosphate leads to a carboxylate shift of D93 accompanied by a shift in the metal position of $Mg_2$ resulting in the "tight" state. In step I, the catalytic base D93 deprotonates the 1' hydroxyl of D-*myo*-inositol-3-phosphate. In step II, D-*myo*-inositol-3-phosphate performs a nucleophilic attack at the β-phosphorus of the CDP-DAG substrate. **c** Presumed penta-coordinated transition state. **d** Release of the reaction product—phosphatidylinositol phosphate and the active site returns to the resting "relaxed" state

| Table 1 Data collection and refinement statistics | | | |
|---|---|---|---|
| | **Apo (PDB: 6H53)** | **CDP-DAG bound (PDB: 6H59)** | **Mn-citrate bound (PDB: 6H5A)** |
| *Data collection* | | | |
| Space group | P $2_1$ $2_1$ $2_1$ | P $2_1$ $2_1$ 2 | P $2_1$ $2_1$ $2_1$ |
| Cell dimensions | | | |
| $a$, $b$, $c$ (Å) | 69.73, 72.90, 102.45 | 97.99, 115.08, 45.47 | 68.07, 77.94, 100.82 |
| α, β, γ (°) | 90.00 | 90.00 | 90.00 |
| Resolution (Å) | 41.28–2.40 (3.00–2.40) | 49.62–1.80 (1.86–1.80) | 40.51–1.88 (1.95–1.88) | 45.79–3.19 (3.19–3.19) |
| $R_{merge}$ (%) | 9.98 (47.83) | 17.50 (114.90) | 14.27 (106.3) | 9.60 (21.56) |
| CC $_{1/2}$ | 0.99 (0.95) | 0.99 (0.68) | 0.99 (0.41) | 0.99 (0.99) |
| $I/\sigma I$ | 12.44 (3.45) | 7.50 (1.70) | 9.52 (1.46) | 22.55 (10.93) |
| Completeness (%) | 99.35 (99.32) | 99.43 (98.61) | 98.87 (97.92) | 98.98 (99.68) |
| Redundancy | 6.60 (6.80) | 12.80 (12.70) | 6.80 (6.70) | 13.20 (13.60) |
| *Refinement* | | | |
| Resolution (Å) | 41.28–2.90 | 49.62–1.80 | 40.51–1.88 |
| No. reflections | 11998 (1164) | 48270 (4693) | 43956 (4281) |
| $R_{work}$/$R_{free}$ | 0.22/0.25 | 0.20/0.23 | 0.20/0.23 |
| No. atoms | | | |
| Protein | 2866 | 3423 | 3502 |
| Ligand/ion | 85 | 291 | 216 |
| Water | 0 | 131 | 126 |
| *B*-factors | | | |
| Protein | 50.88 | 18.49 | 29.77 |
| Ligand/ion | 45.95 | 36.97 | 38.85 |
| Water | - | 27.68 | 32.80 |
| R.m.s. deviations | | | |
| Bond lengths (Å) | 0.009 | 0.010 | 0.009 |
| Bond angles (°) | 0.980 | 1.120 | 1.270 |
| Values in parentheses are for the highest-resolution shell | | | |

DAG bound *M. tuberculosis* PgsA1 dataset was collected at 0.97 Å wavelength and 100 K on beamline I24 at the Diamond Light Source (Oxfordshire, United Kingdom).

**Structure determination, model building, and refinement**. The data were indexed, integrated and scaled using the XDS[28], AIMLESS[29], xia2[30], and DIALS[31] packages. The data resolution cut-off was decided based on $CC_{1/2}$. The apo PgsA1 structure was solved by molecular replacement using PHASER-MR[32–34] and R. *salmoninarum* PIP synthase[19] (PDB: 5D91) as a search model. Prior to molecular replacement the search model was modified by manual truncation of 129 residues from the N-terminus and sequence adaption in Sculptor[32] (Phenix) and manual removal of all bound ligands and solvent. Data for the apo structure was processed to 2.4 Å resolution. However, above the 2.9 Å the R-work and R-free values increased sharply. Consequently, the apo PgsA1 was refined to 2.9 Å. The structure of both substrate-bound PgsA1 was solved using apo *M. tuberculosis* PgsA1 as a search model in PHASER-MR[32,33], followed by refinement in Phenix. refine[33] and iterative building in Coot[35]. Generally, the refinement strategy included individual atomic coordinate and isotropic B factor refinement, bulk solvent correction, occupancy refinement for alternative conformations and bound metal ions. Metal-ligand bonds were restrained. Solvent molecules were added manually. Hydrogen atoms were added to the models in the later stages of refinement. In case of the apo PgsA1 structure, TLS parameters were refined (TLS groups were determined automatically in Phenix.refine). The final structures contained no Ramachandran outliers. Data collection and refinement statistics are given in Table 1. In both structures, some partially ordered lipid molecules that could not be identified by their head groups or other distinctive features were modeled as hydrocarbons of varying carbon number. Disordered hydrocarbon chains of the CDP-DAG ligand were truncated where necessary. Examples of electron density are presented in Supplementary Figs. 5 and 6. All structures were validated using MolProbity[36] and wwPDB Validation Server. All structure figures were prepared with PyMOL[22]. Schematic 2D representation of the protein–ligand complex for Fig. 3 was generated by the LigPlot+ software[37,38]. Protein spatial position in a lipid bilayer was calculated using the PPM server[21].

**Preparation of liposomes and proteoliposomes**. Liposomes were prepared by mixing 16:0–18:1 1-palmitoyl-2-oleoyl-sn-glycero-3-phosphocholine and E. *coli* polar lipid extract in 1:3 proportions (m/m) and addition of 18:1 CDP-DG 1,2-dioleyl-sn-glycero-3-Citidine diphosphate to 30% of a total lipid mass (all lipids were purchased from Avanti Polar Lipids). Chloroform was evaporated under a nitrogen stream and lipid mixture was resuspended in 50 mM Bicine, pH 8.5; 1.5% n-Octyl-β-D-glucopyranoside (w/v) (OG, Anatrace), then incubated for 30 min at room temperature with gentle agitation. Detergent was removed by dialysis overnight using 6–8 kDa cut-off dialysis membrane (ThermoFisher, MA, USA). Proteoliposomes were aliquoted, flash-frozen in liquid nitrogen and stored at −80 °C. Liposomes were thawed on ice and diluted to 10 mg per mL total lipids in buffer containing 50 mM Bicine, pH 8.5, 10 mM $MgCl_2$ and 0.11% Triton X-100. Protein was added to the liposomes in a ratio of 1:80, respectively and gently agitated at room temperature for 20 min. Triton-X was slowly removed by gradual addition of SM2 BioBeads (BioRad, CA, USA). SM2 BioBeads were separated from proteoliposomes by pipetting. Proteoliposomes were concentrated by ultracentrifugation at $190,000 \times g$ for 10 min at 4 °C, aliquoted, flash-frozen in liquid nitrogen and stored until further use (up to 2 weeks).

**Discontinuous activity assay**. The activity assay was performed in proteoliposomes using a set of reagents from Sialytransferase Activity kit (R&D Systems, MN, USA). Assays were performed in triplicate ($n = 3$). Additionally a set of controls was introduced: assay buffer as a blank, and reaction mixture excluding proteoliposomes, providing a control for D-*myo*-inositol-3-phosphate stability and used as the negative control for background extraction. 50 μL reaction mixture contained 50 ng coupling phosphatase 2 (R&D Systems) and assay buffer AB (50 mM Bicine, pH 8.5, 10 mM $MgCl_2$), 1 μg of protein (as measured by Bradford assay) in proteoliposomes doped with CDP-DAG. The reaction was initiated by addition of D-*myo*-inositol-3-phosphate (Cayman Chemical) to final concentration of 0.125 mM and carried out at 37 °C over the time course of 10, 30, 60, 120, and 180 min. Reactions were stopped by addition of 30 μL malachite green reagent A (R&D Systems), 100 μL of $dH_2O$ and 30 μL malachite green reagent B (R&D Systems). After 20 min of color development, optical density of each well was determined using a microplate reader Infinite 200 PRO (Tecan) set at 620 nm.

**Statistics and reproducibility**. Assays were performed in triplicate ($n = 3$) with suitable controls. A phosphate input standard curve was determined using the R&D Systems protocol and assay buffer AB at 37 °C. The phosphate input (pmol) was plotted against the optical density at 620 nm. The derived standard curve equation ($y = 0.0004x + 0.0163$ with $R^2 = 0.997$) was used to calculate phosphate generation: $(OD_{620}\text{-}b) \times a^{-1}$, where $a$ = slope and $b$ = y-intercept, for each assay time point.

**Molecular docking**. Docking was performed using the SwissDock server (http://www.swissdock.ch) using molecular definitions of the substrate D-*myo*-Inositol 3-phosphate obtained from the ZINC database (ZINC02386390)[39]. The docking procedure is based on EADock DSS, including calculation of energies by CHARMM and evaluation and clustering with FACTS as described in ref.[40,41]. Docking was performed against the three structural models apo, CDP-DAG and Mn-citrate bound with all ligands removed and targeted to a box of $10 \times 10 \times 10$ Å$^3$ centered around the Mn-citrate complex binding pocket. Rotatable bonds were allowed to rotate in the ligand. Docking results were visualized by Chimera[42]. In general, ~250 top scoring docking models were clustered into 30–40 clusters of binding poses. Binding modes were inspected manually and plausible substrate binding poses were selected based on binding energy, fitness score and location of key chemical groups. Suggested binding modes were ultimately verified and chosen based on mutagenesis studies and activity assays.

**Reporting summary**. Further information on research design is available in the Nature Research Reporting Summary linked to this article.

## Data availability

The atomic coordinates and structure factors for apo, CDP-DAG bound and Mn-citrate bound *M. tuberculosis* PgsA1 have been deposited in the Protein Data Bank (PDB) under the accession codes 6H53, 6H59, and 6H5A, respectively.

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

## Acknowledgements

We want to thank beamline scientists of Xo6SA at SLS, Paul Scherrer Institute, Villigen, Switzerland and beamline scientists of I24 at the Diamond Light Source, Oxfordshire, United Kingdom for their valuable support. Also we want to thank Geoffrey Masuyer, Hugo Lebrette and Vivek Srinivas for their assistance in X-ray data collection. Additionally, we would like to thank David Drew for access to their Mosquito LCP crystallization robot. This study was supported by the Swedish Research Council (2017-04018) and the Knut and Alice Wallenberg Foundation (Wallenberg Academy Fellows (2012.0233 and 2017.0275)).

## Author contributions

M.B. and M.H. designed the constructs; K.G. cloned, expressed, purified, performed crystallization, data collection and analysis, structure determination and refinement; M.B. and M.H. assisted in structure determination and refinement; K.G. and M.H. planned the activity assays; K.G. performed the activity assays and analysed data; M.H. conceived the study, performed molecular docking experiments, and analysed data; K.G, M.B. and M. H. wrote the manuscript.

## Additional information

**Competing interests:** The authors declare no competing interests.

