## [Peer Review File · Communications Biology]

Reviewers' comments:

Reviewer #1 (Remarks to the Author):

Manuscript#: COMMSBIO-19-0114-T

This manuscript reported several high-resolution x-ray structures of a phosphatidylinositol phosphate synthase from *Mycobacterium tuberculosis*, namely MtPgsA1, that is responsible for the biosynthesis of phosphatidylinositol phosphate. These structures not only agree with previously determined homologous structures on the overall scaffold, but also provide additional high-resolution information, allowing more profound mechanistic insights on substrate binding, coordination of catalytic ions as well as a prediction on how a second substrate ino-P binds to MtPgsA1 in line with current and previous mutagenesis data.

I believe the manuscript can be improved if the following issues are properly addressed.

1. Page 3, Paragraph 2: Citations for the previously determined CDP-AP structures should be added.
2. Page 3, Paragraph 2: It will be helpful to clarify if MtPgsA1 assembles as dimers in biological membrane with each protomer capable of catalyzing PI synthesis.
3. Page 4, Paragraph 1: *Renibacterium salmoninarum* was mis-spelt as "Renibacterium salmonarium".
4. Figure 1, Panels C and E: I suggest showing Fo-Fc omit maps for citrate instead of 2Fo-Fc maps.
5. Page 4, Paragraph 1 from bottom: I suggest the authors to further discuss about the interesting observation of differently positioned Mn-citrate complex in each protomer, on topics related to if either or both protomers represent a native-relevant apo state; if crystal packing contributed to asymmetric protomers; and if citrate is present in *Mycobacterium tuberculosis*. If citrate were not a relevant salt, it would be ideal to attempt crystallization with citrate replaced by a similar buffer, to investigate how Mn-citrate complex influence Mn^{1/2} binding.
6. Figure 2: I suggest a different choice of color for Mg, to be distinguishable from the green Fo-Fc map.
7. Page 6, Paragraph 1: Figure 3 was called before Figure 2.
8. Page 8, Paragraph 1 from bottom. I suggest the authors to perform molecular dynamic simulation studies to validate the results of docking.

Reviewer #2 (Remarks to the Author):

The manuscript reports three crystal structures of a phosphatidylinositol phosphate synthesizing enzyme from a dangerous human pathogen, *Mycobacterium tuberculosis*. Development of an effective pharmaceutical disrupting this enzyme is of great interest since others identified it to be essential for growth and proliferation of *M. tuberculosis*.

The structure in complex with one of the two substrates shows the details of its binding mode. A combination of molecular docking supported by mutagenesis and functional assays enabled placing the second substrate in the active site in two possible binding modes. The high resolution of the reported structures with insights into the substrates binding warrants usefulness of this work for structure-based drug development.

There are only a few other examples of structures of the CDP-alcohol phosphotransferases. The reviewer assumes that these structures are very similar as there is a little comparison in the manuscripts. Nevertheless, the high resolution of presented here structures enables an interesting discussion about the possible mechanism of catalysis and the role of the 310 helix motif, which may be relevant for the whole family of these lipid synthesizing enzymes.

-Authors state that in the metal-free structure two important Asp residues are disordered. Do the authors notice any signs of radiation damage in the processed data? Are any other Asp/Glu residues affected away from the active site (e.g., show negative density in Fo-Fc maps at 2.5 rmsd)? Did the crystal in the presence MnCl₂ received lower radiation dose and thus the Asp residues are resolved?

-Table I: Please specify what is the common criterium for determining the high-resolution cutoff for the reported datasets? The reported I/sigI and CC1/2 for the high-resolution shell in APO data is 3.45 and 94.80 respectively suggesting that not all useful data might have been included in the refinement and consequently could contribute to the inability to resolve the metal binding site.

-Table I: Rpim, CC1/2, anomalous correlation and completeness are likely given as % value not absolute values. Please specify in the table.

-p.4: "The structure reveals a di-nuclear metal binding site coordinated by D68, D71, D89 and D93 of the conserved sequence motif. In chain A, the positions are well defined with two clear peaks in this area (fig. 1, B)." The reviewer sees 4 anomalous difference peaks indicating metal binding sites in fig. 1, B. Please explain why "only" di-nuclear metal-binding site was concluded.

-p. 8: "This had been previously speculated" – a reference missing.

-p. 9: "Cocrystallization and soaking attempts with Ino-P were unsuccessful, as previously observed and in line with the low binding affinity for this substrate (18)."

a. The ref. 18 Sciara et al 2014 does not mention cocrystallization or soaking attempts with Ino-P. Neither it mentions the binding affinity of Ino-P. Please correct the reference.

b. Instead, the Clarke et al (2015) does mention 243 μM K_m of Ino-P, however, is it so much lower than the 60 μM K_m of CDP-DAG, which was successfully trapped in the crystal structure, to blame the affinity for unsuccessful cocrystallization with Ino-P? Authors convincingly argue (and in line with others) that the Ino-P should bind to a positively charged pocket, which is occupied either by sulfate or citrate in the presented structures. Isn't it more likely that the presence of high concentrations of sulfate and citrate in the crystallization conditions prevented the binding of the Ino-P?

-Figure S1: For clarity add information that the black line indicates the $i + 3 \rightarrow i$ hydrogen bond defining 310 helix motif. Perhaps using a distinct color of the portion of the helix cartoon corresponding to the motif would improve the clarity of the figure.

Reviewer #3 (Remarks to the Author):

The manuscript by Grave et al reports the structure of *Mycobacterium tuberculosis* phosphatidylinositol synthase (PgsA1) in three forms (apo, citrate-bound and CDP-DAG bound). The protein is important as it has been suggested as a potential drug target for small molecules that will kill this pathogen. Structures of related enzymes have been solved previously. Indeed, Clarke et al solved the structure of the enzyme from *Renibacterium salmoninarum* with 40% sequence identity to the Mtb protein and used this to make a homology model of the Mtb protein. This was then used as the basis to probe the function of the protein through site-directed mutagenesis. In the study reported here, Grave et al have obtained higher resolution data enabling them to better define the substrate and metal binding site. They have also fortuitously solved the structure of the protein with a citrate molecule, which they claim helps them understand how the natural substrate should bind. Finally, they carry out docking studies with the natural substrate and test the resultant poses through site-directed

mutagenesis and activity assays.

The structure looks reasonable and the electron density that they show looks excellent and consistent with the reported resolution.

Overall, I found some of the descriptions confusing and I found it difficult to relate what was being said to the figures. For example, why are there 4 sites marked in Figure 1b? This is a bi-nuclear site.

My major comments refer to how they think the inositol phosphate binds as this is significant to the mechanism.

Although in the introduction it was mentioned that the position of the citrate gave some clue about the ino-P binding site, this was not really further discussed in the results section and instead the positions of the sulphates were given more prominence. How does the position of the citrate relate to the position of the CDP-DAG or to the modelled ino-P? An additional figure would have been useful.

It is very hard to understand how the ino-P binds in Figure 3 and hence assess the significance of the docking poses. There is very little information about the docking in the methods – how have the poses that they have used been selected etc? I would have also like to see some discussion of the mutations and analysis made by Clarke et al.

Minor comments:

Is there any significance of the phospholipid binding sites that between the protein chains that is described in the manuscript?

It is very difficult to see the stick representations in Fig 2d.

Reviewer #1 (Remarks to the Author):

1. Page 3, Paragraph 2: Citations for the previously determined CDP-AP structures should be added.

Citations for the previously determined CDP-AP structures were added (Nogly et al.; Sciara et al.; Clarke et al.)

2. Page 3, Paragraph 2: It will be helpful to clarify if MtPgsA1 assembles as dimers in biological membrane with each protomer capable of catalyzing PI synthesis.

Based on currently available structural models and mutational studies of several members of the CDP-AP family, we know that both protein chains of the homodimer constitute the substrate-binding site. This implies that a dimer must be a minimal functional oligomeric state for this type of enzymes. Based on currently available data, it is not possible to conclude if both protomers are capable of simultaneous catalysis, if catalysis happens in alternating fashion, or if only one protomer is active. This is indeed the case for most homomultimeric enzymes. To our knowledge, the general assumption is that monomers are catalytically equivalent in these cases, but it cannot be explicitly concluded.

3. Page 4, Paragraph 1: *Renibacterium salmoninarum* was mis-spelt as “*Renibacterium salmonarium*”.

This has been corrected.

4. Figure 1, Panels C and E: I suggest showing Fo-Fc omit maps for citrate instead of 2Fo-Fc maps.

This has been changed according to the reviewer’s suggestion, now an Fo-Fc omit map is displayed around bound citrate molecules, contoured at 3 sigma and shown as a grey mesh. In order to increase the color contrast, we have changed the color representation for the citrate ions from green to orange. Similarly, panel F was modified: The citratechelated Mn ions were hidden for clarity, APO protein shown in green (previously in gray) and Mn-bound – in blue (previously in green). The Figure legend has been modified accordingly.

5. Page 4, Paragraph 1 from bottom: I suggest the authors to further discuss about the interesting observation of differently positioned Mn-citrate complex in each protomer, on topics related to if either or both protomers represent a native-relevant apo state; if crystal packing contributed to asymmetric protomers; and if citrate is present in *Mycobacterium tuberculosis*. If citrate were not a relevant salt, it would be ideal to attempt crystallization with citrate replaced by a similar buffer, to investigate how Mncitrate complex influence Mn^{1/2} binding.

As suggested by the referee, we have added further discussion about the citrate complexes and the observed differences in the protomers. Regarding the question whether the Mn-citrate complex structures represent a native-relevant apo-state we feel that the actual apo structure (also presented in this paper) remains a better model for the apo state and should preferentially be used as the reference for this. Citrate is present at high concentration (100 mM) in the crystallization solution and thus the Mn-citrate complex is most likely not physiologically relevant. However, the CDP-DAG structure is crystallized in the absence of citrate and show the same metal binding positions in the catalytic site. For this reason, we do not believe that the additional Mn-citrate complex significantly influences the protein-coordinated metal sites.

6. Figure 2: I suggest a different choice of color for Mg, to be distinguishable from the green Fo-Fc map.

We have updated this figure according to the suggestions by the referee. Magnesium ions are now shown in magenta. Additionally, in order to increase color contrast in figure 2D, the CDP-DAG is now shown in cyan. The Fo-Fc omit map is shown at 2.5 sigma and the figure legend is modified accordingly.

7. Page 6, Paragraph 1: Figure 3 was called before Figure 2.

This has been corrected.

8. Page 8, Paragraph 1 from bottom. I suggest the authors to perform molecular dynamic simulation studies to validate the results of docking.

As stated, the docking was performed to investigate if the binding site defined by the citrate complex would support a plausible binding mode for the ino-P substrate, which docking suggested. The proposed binding modes (illustrated by one representative model for each cluster of structures) were validated using mutagenesis studies and activity assays. We choose to validate the docking results using mutagenesis rather than molecular dynamics as it has the added benefit of providing information on activity and mechanism. We agree with the reviewer that it would certainly be interesting to further investigate the details of the interaction in a study focused on the ino-P binding using additional theoretical and experimental methods, in particular NMR. However, we feel that this would be outside the scope of the present study. We have further detailed the docking methods and process (see also reply to reviewer #3)

Reviewer #2 (Remarks to the Author):

-Authors state that in the metal-free structure two important Asp residues are disordered. Do the authors notice any signs of radiation damage in the processed data? Are any other Asp/Glu residues affected away from the active site (e.g., show negative density in Fo-Fc maps at 2.5 rmsd)? Did the crystal in the presence MnCl₂ received lower radiation dose and thus the Asp residues are resolved?

No, we do not notice signs of radiation damage, such as decarboxylation of aspartate or glutamate residues at the metal site or elsewhere in the structure. Both crystals (APO and Mn-containing) were approximately the same size/volume and collected at the same beamline. The data collection parameters for the native datasets were identical. In fact, the Mn-containing crystal was exposed to a higher total radiation dose, because an anomalous dataset had been collected on the same crystal first. We do not see any signs of de-carboxylation of metal-chelating aspartates (or solvent-exposed carboxylates) in the Mn-containing structure. For these reasons we are convinced that the lack of density in the metal-free structure is due to disorder of the metal ligands and not due to radiation damage.

-Table I: Please specify what is the common criterium for determining the highresolution cutoff for the reported datasets? The reported I/sigI and CC1/2 for the highresolution shell in APO data is 3.45 and 94.80 respectively suggesting that not all useful data might have been included in the refinement and consequently could contribute to the inability to resolve the metal binding site.

The resolution cut-off for the high-resolution native datasets was estimated by the CC1/2 value. In the case of the APO dataset, even though the data quality parameters potentially indicated that the data extended to beyond 2.9 Å it quickly decreased in quality and the refinement behavior suggested that the data above 2.9 Å was not useful (as judged by monitoring R-work and R-free per resolution bin and map quality during the refinement process). For this reason, we decided to be conservative with the resolution cutoff in this case. We have included a clarification in “Materials and Methods” section “Structure determination and model building”.

-Table I: Rpim, CC1/2, anomalous correlation and completeness are likely given as % value not absolute values. Please specify in the table.

This has been corrected.

-p.4: “The structure reveals a di-nuclear metal binding site coordinated by D68, D71, D89 and D93 of the conserved sequence motif. In chain A, the positions are well defined with two clear peaks in this area (fig. 1, B).” The reviewer sees 4 anomalous difference peaks indicating metal binding sites in fig. 1, B. Please explain why “only” di-nuclear metal-binding site was concluded.

The additional two anomalous peaks stem from the Mn-citrate complex and not proteincoordinated metal ions. The paragraph has been edited to avoid this potential misunderstanding.

-p. 8: “This had been previously speculated” – a reference missing.

The relevant references have been added (Sciara et al.; Clarke et al.).

-p. 9: “Cocrystallization and soaking attempts with Ino-P were unsuccessful, as previously observed and in line with the low binding affinity for this substrate (18).”

a. The ref. 18 Sciara et al 2014 does not mention cocrystallization or soaking attempts with Ino-P. Neither it mentions the binding affinity of Ino-P. Please correct the reference.

We thank the referee for pointing out this mistake, the reference has been exchanged for the correct one (Clarke et al.).

b. Instead, the Clarke et al (2015) does mention 243 μM K_m of Ino-P, however, is it so much lower than the 60 μM K_m of CDP-DAG, which was successfully trapped in the crystal structure, to blame the affinity for unsuccessful cocrystallization with Ino-P? Authors convincingly argue (and in line with others) that the Ino-P should bind to a positively charged pocket, which is occupied either by sulfate or citrate in the presented structures. Isn't it more likely that the presence of high concentrations of sulfate and citrate in the crystallization conditions prevented the binding of the Ino-P?

The referee is indeed correct, we believe that the low affinity of the Ino-P towards the Mt PgsA1, compared to CDP-DAG, as well as competition with sulphate/citrate ions for the binding site is the reason for the lack of success in the co-crystallization attempts with this substrate. A clarification has been added to the manuscript main text in the same paragraph.

-Figure S1: For clarity add information that the black line indicates the $i + 3 \rightarrow i$ hydrogen bond defining 310 helix motif. Perhaps using a distinct color of the portion of the helix cartoon corresponding to the motif would improve the clarity of the figure.

Figure S1 and its legend have been modified according to the reviewers suggestions. Additionally, we have shown the hydrogen bond distance between M69 and L66 and labeled transmembrane helix 1 (TM1).

Reviewer #3 (Remarks to the Author):

Overall, I found some of the descriptions confusing and I found it difficult to relate what was being said to the figures. For example, why are there 4 sites marked in Figure 1b? This is a bi-nuclear site.

The additional two anomalous peaks stem from the Mn-citrate complex and not proteincoordinated metal ions. The paragraph has been edited to avoid this potential misunderstanding.

Although in the introduction it was mentioned that the position of the citrate gave some clue about the ino-P binding site, this was not really further discussed in the results section and instead the positions of the sulphates were given more prominence. How does the position of the citrate relate to the position of the CDP-DAG or to the modelled ino-P? An additional figure would have been useful.

We agree with the referee that this can be more clearly shown. Figure 4 has now been remade to better illustrate the proposed binding modes for ino-P in relation to the sulfates and citrate binding positions. The figure legend and corresponding results section has also been modified accordingly.

It is very hard to understand how the ino-P binds in Figure 3 and hence assess the significance of the docking poses. There is very little information about the docking in the methods – how have the poses that they have used been selected etc? I would have also like to see some discussion of the mutations and analysis made by Clarke et al.

We have further detailed the docking methods, process and reasoning regarding selection of the poses. Also, figure 4 has been modified to aid visual evaluation of the representative ino-P binding modes (see above). In addition, the relevant mutations, described by Clarke et al. have been further discussed.

Minor comments:

Is there any significance of the phospholipid binding sites that between the protein chains that is described in the manuscript?

The phospholipid binding sites discussed in the manuscript do not appear to be significant in terms of the mechanism. However, given the high degree of order for these lipids, they appear tightly bound and relevant for the structural description of the enzyme.

It is very difficult to see the stick representations in Fig 2d.

The color for the CDP-DAG has been changed to increase clarity

REVIEWERS' COMMENTS:

Reviewer #1 (Remarks to the Author):

I believe that the authors have addresses my concerns to satisfaction, and thus recommend the manuscript for publication.

Reviewer #3 (Remarks to the Author):

All comments seem to have been addressed satisfactorily. Figure 4 is much improved.